# Hidden Morphotypes and Homologous Series in Phenotype Variations in the Colonial Hydroids *Dynamena pumila*, *Diphasia fallax*, and *Abietinaria abietina* (Hydrozoa, Leptothecata)

**Nikolay N. Marfenin**

Department of Invertebrate Zoology, Lomonosov Moscow State University, 119991 Moscow, Russia; marf47@mail.ru; Tel.: +7-(916)-8321866

**Abstract:** The intraorganismal variability of the shoot modules of three species of hydroids was studied to determine the degree of similarity between them. The strict form of the internodes (modules) of the shoots is repeated many times, which is useful for the study of intraorganismal variability. Against the general background of the high stability of the shape of the internodes, we found significant deviations from the norm. Some resemble the structure of the internodes in other genera of the same family. Their morphogenesis is different from that characteristic of the studied species. Most of the anomalies were characterized by stable forms and low frequencies of occurrence (<0.2%). After the appearance of abnormal internodes, normal ones were found to usually re-form. Thus, it is doubtful that the anomalies were caused by mutations. There is also no reason to believe that the anomalies were caused by environmental factors, since they always formed singly along with normal shoot modules of the same modular organism. In *Dynamena pumila*, *Diphasia fallax*, and *Abietinaria abietina*, the composition of their morphovariations was found to be similar, and their frequencies were comparable, which confirms the assumption that several latent phenotypes can be formed indeterminately based on one genotype. The study was conducted on samples of >20,000 internodes of each of the three species.

**Keywords:** phenotypic plasticity; morphological abnormalities; ecology; evolution

## 1. Introduction

Phenotypic plasticity or variability has been described in many species [1–4]. Researchers' interest in this topic mainly focuses on the influence of environmental factors on the form of an organism [3,5,6] and on the evolutionary significance of phenotypic plasticity [7].

Additionally, insufficient attention has been paid to the ability of a species to exist in several forms. We know from many examples that an organism can indeed have many forms, and these differences between the phenotypes of one species, and even one organism, can amaze us with the degree of difference between them, such as the morphotypes of a caterpillar and a butterfly. Nevertheless, many articles are limited to phenotypic plasticity only, i.e., the description of the morphological response to the impact of environmental factors. However, alternative phenotypes may be out of touch with the environment. They may appear infrequently and without any connection to life cycle stages, sex differences, age changes, or environmental influences. What is the significance of the ability of morphogenesis to produce forms that are not characteristic of the species? What are "memories of the past" or "embryos of the future"? In this article, we describe the initial results of studying such a phenomenon, namely, morphogenetic polyvariance using the example of modular organisms.

Morphogenetic polyvariance is the presence of several stable phenotypes in a species that are not associated with either sexual dimorphism, life cycle stages, or the living conditions of the organism [8].

Morphogenetic polyvariance was first discovered in the colonial hydroid *Dynamena pumila* (L., 1758) [9]. In this species, the hydranths on the shoots are strictly opposite in two rows [10–12] (Figure 1). Any deviation from this pattern is easily recognizable. This is why *D. pumila* was chosen to study morphological variability. The repeated repetition of the stereotypical form of the internode (=module) of the shoot in the colonial organism made it possible to study the variability not only in the population, but also within the organism. The so-called "colony" in hydroids is not a population, but a decentralized organism in which the body consists of many repeating parts of several varieties, and there are no controls. Additionally, the single body of a colonial organism has a distribution system that physiologically unites all its parts [13–15].

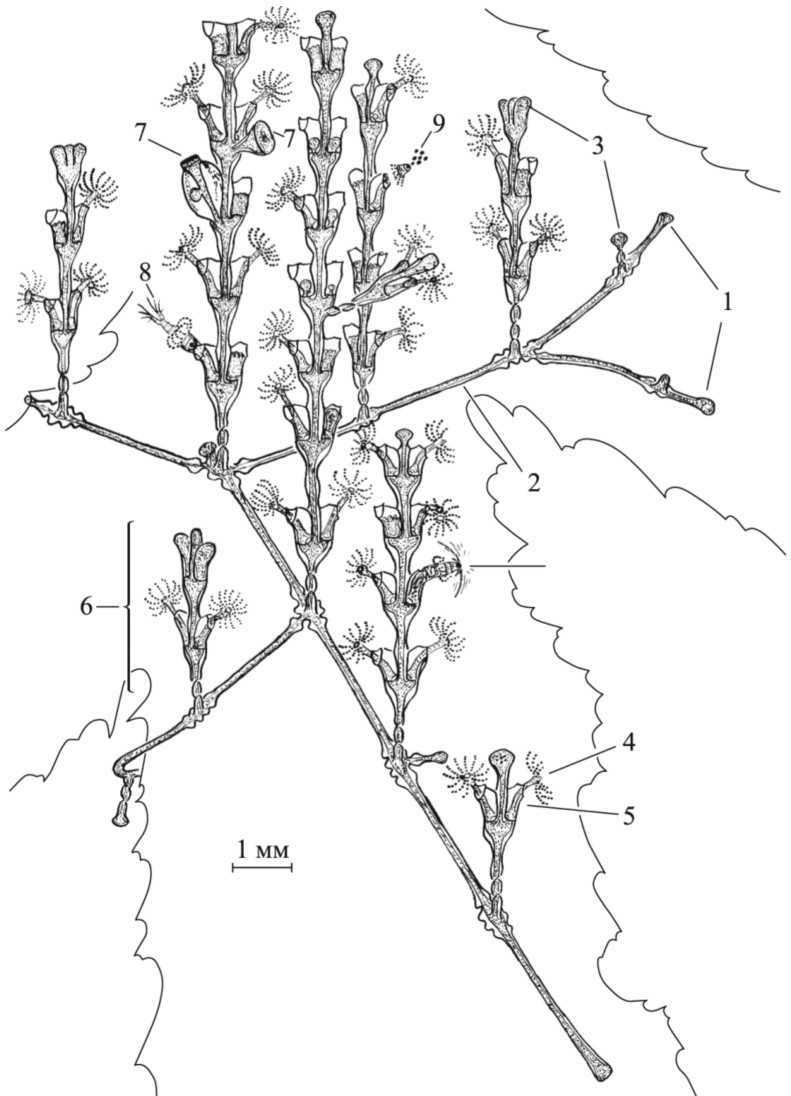

**Figure 1.** Schematic drawing of a small *D. pumila* colony (adapted from [8]). *Designations*: 1—growth tips of stolons; 2—stolon; 3—growth tips of shoots; 4—hydranth; 5—hydrotheca; 6—shoot (or stem); 7—gonotheca; 8—hydranth ingesting prey; 9—hydranth squeezing out undigested food residues.

It was found that, along with the standard internodes characteristic of the shoots of *D. pumila*, modules of a different shape are formed from time to time. Gradually, it was possible to compile a description of the discovered aberrant modules and other deviations from the norm of the structure of the shoot [16].

Usually, anomalous internodes are formed on shoots after perfectly normal ones, and, quite often, normal ones are formed again after anomalous modules. This means that

there can be two or more phenotypes in one organism. When studying many samples of *D. pumila* shoots, it was found that anomalous modules appear with certain frequencies, which can vary, but not within significant limits.

The above indicates the difference between morphogenetic polyvariance and other types of variability. At least six variants of the morphological variability of an organism are known:

1.  Sexual dimorphism.
2.  Polymorphism: The presence of several variants of the structure of zooids in the colony (in Hydrozoa and Bryozoa).
3.  Sequentialchange in phenotypes during the life cycle: For example, in the classes Hydrozoa and Scyphozoa, the formation of a medusoid stage after a polypoid one is possible; in many parasitic organisms, a change in phenotype occurs during the transition from one stage of the life cycle to the next; in the Insecta class, individual development usually includes metamorphosis. There are many such examples. Moreover, the ontogenesis of any species can be represented by a series of phenotypes.
4.  Intrapopulation morphological variability: This reflects the breadth of variations in morphological traits accumulated in the gene pool of the species during crossing and exchange of genetic information.
5.  Phenotypic plasticity: A change in the shape of an organism or its individual parts in response to changes in environmental parameters. This phenomenon has recently attracted significant attention from biologists [3,17,18].
6.  Deformities: Deviations of the form and structure of the body from the norm as a result of genetic mutations. Depending on the nature of the mutations, deformities can be inherited or appear in one generation. In the latter case, these are called somatic mutations.

Morphogenetic polyvariance in the form of abnormal shoot modules arising from time to time does not correspond to any of the above forms of morphological variability of an organism [8]. We assumed that morphogenetic polyvariance is a manifestation of "hidden" phenotypes, i.e., that the body has enough genetic resources to ensure morphogenesis along several pathways, some of which are "canonical", for example, sexual dimorphism or changes in forms during individual development, whereas others are "spare".

For most "reserve" phenotypes, it is known that they appear under certain external influences. This is "phenotypic plasticity" [3]. The influence of environmental factors in such cases is natural and affects the morphogenesis of all (or most) objects dependent on them. In the colonial hydroid, *D. pumila*, internodal anomalies are always single. Of all the shoots, only a thousandth or ten thousandth part of the simultaneously developing modules turns out to be aberrant. What is the influence of the environment here?

All of this enhances interest in morphogenetic polyvariance and encourages the study of this phenomenon in other species.

This article describes the morphogenetic polyvariance in three types of hydroids, characterized by a two-row arrangement of the hydrothecae.

The article focuses on the following questions:

*   To what extent is the phenomenon of morphogenetic polyvariance inherent in other species, and not only in *D. pumila*?
*   To what extent are module anomalies similar in species belonging to different genera of the same family?
*   Do similar anomalies differ among themselves in different species in terms of their frequency of occurrence?

This study was carried out on three species from the Sertulariidae family: *Diphasia fallax* (Johnston, 1847), *Dynamena pumila* (Linnaeus, 1758), and *Abietinaria abietina* (Linnaeus, 1758). For the convenience of analyzing the degree of similarity between the three species, the following sections of the article present both new data on *D. fallax, D. pumila*, and *A. abietina*, as well as previously published [8] data on *D. pumila* with appropriate links.

The hydrothecae in *D. fallax* are located opposite each other in the same way as those in *D. pumila* [19], whereas in *A. abietina,* the hydrothecae are located alternately, i.e., with some shifts relative to each other. In other words, the comparison was made between species differing in the degree of similarity in the structure of shoots.

## 2. Material and Methods

### 2.1. Sample Collection

The samples of *D. fallax, D. pumila*, and *A. abietina* used in this study were collected in different years in the same geographical location (66°34′ N, 33°08′ E), i.e., in Velikaya Salma of the Kandalaksha Bay of the White Sea in the immediate vicinity of the White Sea Biological Station of Moscow State University (Table 1). *D. fallax* and *A. abietina* specimens were collected by divers in 2000 and 2001 at a depth of 10–15 m from a shell rock. These species live on a strong tidal current, the maximum speed of which reaches 1 m/s.

**Table 1.** The number of studied shoot internodes and the number of colonies of *Dynamena pumila* (Linnaeus, 1758), *Diphasia fallax* (Johnston, 1847), and *Abietinaria abietina* (Linnaeus, 1758). The number of morphovariations, including the most common ones. Cumulative frequency of occurrence of shoot morphovariations.

| Species | Total Number of Internodes | Number of Samples | Total Number of Morphovariations | Number of Morphovariations Dominating in Occurrence (>0.1%) | Cumulative Frequency (%) of Occurrence of Morphovariations |
|---|---|---|---|---|---|
| *A. abietina* | 20,395 | 2 | 39 | 5 | 1.59 |
| *D. pumila* | 230,700 | 23 | 32 | 3 | 1.45 |
| *D. fallax* | 21,737 | 2 | 26 | 9 | 1.00 |

A total of 25 samples of *D. pumila* were collected in 1991–1994, also near the White Sea Biological Station, Lomonosov Moscow State University at the lower boundary of the littoral zone, mainly on the Yeremeevsky threshold, where algae (*Fucus serratus* and *Ascophillum nodosum*) overgrow colonies of this species.

### 2.2. Sample Preservation

The samples of hydroids were preserved in 70° ethanol. This method of fixation did not affect the results of the study because it is based on taking into account the shape of the chitinoid perisarc, which is not affected by conventional fixatives.

### 2.3. Registration of Abnormal Morphovariations

The shoots were carefully examined internode by internode under a stereomicroscope LZOS, MBS-9 (at a magnification of no more than ×30), and any deviations from the normal shape or position of the partswerenoted. The results were entered into a spreadsheet and illustrated with biological drawings that clearly indicate the location of the hydrothecae, lateral branches, and other parts of the perisarcal cover of the shoots. Typical anomalies were photographed, but it was not always possible to present the characteristic features of anomalies in photographs in comparison with biological drawings.

Previously compiled tables were used to classify anomalies [16]. If necessary, a description of new anomalies was added.

### 2.4. The Anomaly Number Consists of Three Parts

Roman numerals indicate the group to which this anomaly is assigned. Arabic numerals after a dash indicate the serial number of the anomaly in this group. Arabic numbers in parentheses indicate the anomaly number in the first published classification [9].

Example: "IV-1(9)" represents the following: group IV: deviations from the norm of the hydrotheca structure; 1 (in the classification of 1995): hydrotheca on hydrotheca; (9): number in the 1975 classification.

### 2.5. Terminology Used in the Article

Shoot: Part of a colony with hydranths ascending from the substrate or a stalk extending from the stolon with lateral branches on which hydranths are located.

Stalk of the shoot: The central part of the shoot, from which lateral branches can extend.

Shoot module: An internode, including a segment of the stalk with hydranths extending from it, which is formed in one cycle of morphogenesis (Figure 2).

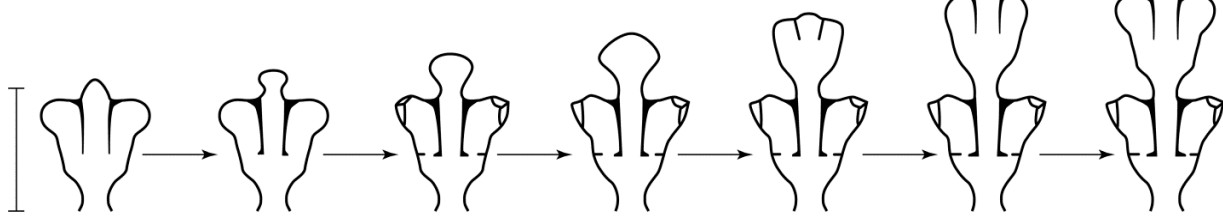

**Figure 2.** Morphogenetic shoot cycle of *D. pumila*. Scale bar = 1 mm.

Shoot growth apex: the apical part of the shoot coenosarc undergoing cyclic morphogenesis, in which intercalary elongation of the coenosarc occurs in the proximal part.

Cyclic morphogenesis (Figure 2): A sequential change in shape, culminating in the initial stage, during which one module is formed. Cyclic morphogenesis is the basis of the plasticity of a modular body, i.e., the ability to adapt to environmental conditions [16].

Shoot shape anomalies: Any deviation from the species-specific form of shoots and hydrothecus; unusual locations of the hydrotheca, lateral branches, and gonothecus; and deformities.

Morphological variations (or morphovariations or morphovarieties): Certain repeatedly occurring variants of the unusual structure of the shoot module and the location of the lateral branches of the stolons and gonothecae.

Morphotypes: Special viable morphovarieties, sometimes repeated several times in a row in the process of cyclic morphogenesis, similar to other species and genera.

Spare morphogenesis refers to alternative programs for the formation of body parts and organs that are usually implemented rarely, but it is assumed that, under some conditions, they can become the main programs for the individual development of a species.

To determine the frequency of occurrence of each anomaly (AF—anomaly frequency), the ratio between the number of each type of anomaly and the total number of hydrothecae studied was calculated.

To determine the cumulative frequency of occurrence of all the anomalies (CF—cumulative frequency of anomalies), the ratio between the sum of anomalies of all the varieties and the number of hydrothecae studied is calculated.

To determine the relative frequency of occurrence of anomalies (RF—relative frequency of anomalies), the percentage ratio of the number of modules of a certain anomaly to the total number of all anomalous modules in the sample was calculated.

### 2.6. Standard Sample Size

As shown earlier for *D. pumila*, a representative sample for calculating the frequency of occurrence of the most common morphovariations should be at least 3000 internodes [8,16]. In this study, the standard sample included 10,000 internodes (Table 2).

**Table 2.** The 23 main morphovarietiesof shoots of D. pumila in descending order of frequency of occurrence in a total sample of 230,700 shoot modules (White Sea).

| No. According to the Classification of Anomalies in *D. pumila* 1995 (1975) | Short Name for *D. pumila* Shoot Anomalies | Sum of Anomalies in the Total Sample | Frequency of Occurrence of Anomalies in the Total Sample (%) |
|---|---|---|---|
| I-20(23) | Hydrothecae in two rows with an offset relative to each other | 1264 | 0.548 |
| I-40(3) | Trunk extension | 551 | 0.239 |
| I-60(24) | Single row of hydrothecae | 374 | 0.162 |
| V-50(20) | A branch from a hydrotheca | 139 | 0.060 |
| I-50(20) | Long constriction between pairs of hydrothecae | 138 | 0.060 |
| I-62(25) | Single row of hydrothecae to double row | 123 | 0.053 |
| V-1(15) | Branch perpendicular to frontal plane of a shoot | 102 | 0.044 |
| IV-1(9) | Hydrotheca on hydrotheca | 99 | 0.043 |
| IV-35+I2 | Barrel deformation | 94 | 0.041 |
| (IV5+IV6)(9b) | One hydrotheca for two internodes | 84 | 0.036 |
| VII-20(10) | Hydrotheca's stolon | 58 | 0.025 |
| I-1(1) | Trunk bend | 52 | 0.023 |
| II10(8) | Two adjacent hydrothecae | 49 | 0.021 |
| 1–10(4) | Rotation of the plane of the colony around its own axis | 46 | 0.020 |
| II-40 | Ugly growth tip of a stem | 35 | 0.015 |
| IV-30 | Ugly hydrotheca | 33 | 0.014 |
| II-1(12) | Hydranth at the top of the stem | 30 | 0.013 |
| V-60(22) | Branch of a stem from the gonotheca | 29 | 0.013 |
| I-82(18) | Three trunks growing from the top of the shoot | 15 | 0.007 |
| IV-20a(13) | A single hydranth on the trunk instead of a branch; the hydranth grows from a bud at the base of the internode and is devoid of a growth tip | 12 | 0.005 |
| VII-1(17) | The tip of the shoot is reborn into a stolon | 8 | 0.003 |
| I-70(27) | Three-row arrangement of hydrothecae | 7 | 0.003 |
| VII-10 | The stolon grows from the trunk of the shoot | 5 | 0.002 |

Alternative morphotypes are highlighted in red.

To identify the main shoot anomalies in *D. fallax* and *A. abietina*, samples of 20–21 thousand shoot modules were processed (Table 1).Theresults were then comparedwithdata obtained from a total sample (230,700) of *D. pumila* shoot internodes (from the White Sea only) [16]. The study of morphovariations in *D. pumila* shoots has continued for several years. Therefore, the available sample of *D. pumila* shoot modules is more than ten times larger than that for each of the other two species (Table 1).

*2.7. Similarities and Differences between Compared Species*

Three species of colonial hydroids, namely, *D. fallax*, *D. pumila*, and *A. abietina*, are united by several common features:

- A two-row arrangement of the hydrothecae on the truck and branches;
- Branches depart from the truck in the frontal plane of the stem;
- The dimensions of the hydrothecae on the stalk and branches are the same;
- Pitcher-shaped hydrothecae with an expanded base;
- Hydrothecae in the lower part adjoin on one side to the perisarc of the stalk.

Differences between species are mainly expressed in the following features:

○ An opposite and alternate arrangement of the hydrothecae; namely, in *D. pumila* and *D. fallax*, the hydrothecae are located strictly opposite on the stem and branches, and, in *A. abietina*, the arrangement of the hydrothecae is alternate, i.e., they are displaced along the axis of the stem relative to each other, being in the plane of the shoot;

○ The number of valves in the operculum (the valves cover the mouth of the hydrotheca). In *D. pumila*, the operculum of the hydrotheca consists of two valves of thin perisarc, whereas in the other two species there is only one valve;

○ In *D. fallax*, the top of a large stem often transforms into a curved stolon-like tendril; the other two species do not normally have this.

In addition, there are clear differences in the structure of the gonothecae of the compared species, but we did not take these into account.

## 3. Results

### 3.1. Brief Classification of Morphovariations According to the Example of Dynamena pumila (Linnaeus, 1758)

Before comparing the three biological species of hydroids according to the sets and the frequency of occurrence of morphovariations in shoot internodes, it worth presenting the classification of morphovariations in *D. pumila* that was developed earlier [9,16], taking into account new data on the frequency of occurrence of various morphovariations in a large sample from the White Sea.

Among 230,700 studied internodes of the shoots of *D. pumila*, 47 major morphological anomalies were found, not counting their varieties [16]). Anomalous *D. pumila* shoot modules are rare. Only among hundreds of internodes of the shoots were there single deviations from the norm. All the morphovariations of *D. pumila* were divided into four groups [9,16] (Figures 3 and 4):

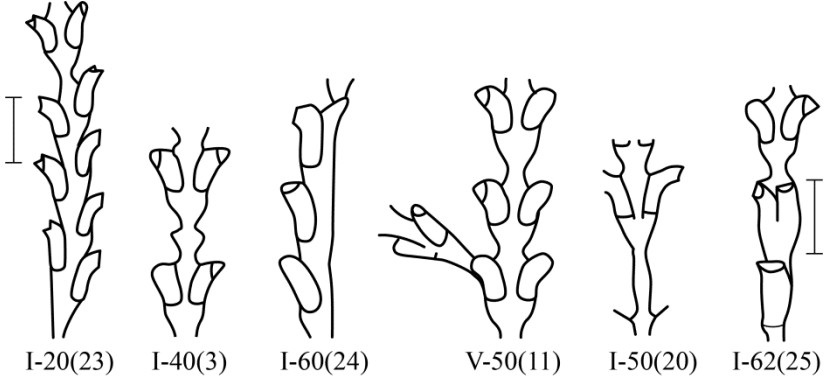

I-20(23)    I-40(3)    I-60(24)    V-50(11)    I-50(20)    I-62(25)

**Figure 3.** The predominant morphovarieties in the structure of the shoot *D. pumila*. Scale bars = 1 mm. The morphovariety numbers correspond to the complete classification (according to [9,16]).

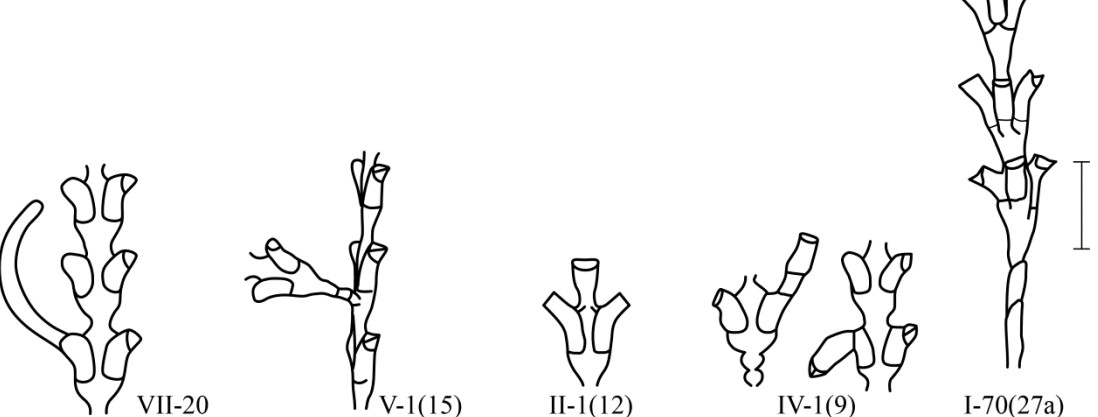

VII-20    V-1(15)    II-1(12)    IV-1(9)    I-70(27a)

**Figure 4.** Examples of morphovarieties in the structure of the *D. pumila* shoot that are less common. Scale bar = 1 mm. The morphovariety numberscorrespond to the complete classification (according to [9,16]).

The first group of morphovariations is due to errors in the restoration of shoot growth. This is indicated by the shape of the anomalies and their position in the shoot of the colony.

The second group of morphovariations can be called errors of morphogenesis without an explicit connection to the suspension of growth.

The third group of morphovariations are stable viable forms that are not characteristic of this species, but rather characteristic of another genus.

The fourth group of morphovariations, obvious deformities of a non-permanent form, are usually not viable.

The first group of morphovariations mainly includes anomalies that occur when growth resumes after it has stopped. As a rule, shoot growth stops in *D. pumila* after the formation of the next internode is completed. If the growth of the shoot resumes after some time, then the emerging tip of the coenosarc of the trunk is usually located strictly along the axis of the shoot, but sometimes it dissolves the perisarc from the side, and then the next internode is formed at an angle to the axis of the shoot. It is possible to change the plane of the shoot module after growth resumes {I-10(4)}. The suspension of shoot growth can often be judged by the color of the perisarc. In this location, there is a border between the dark perisarc located below and the light one located above, i.e., between the old and the young persiarcs. There are many similar examples in early spring, when the growth of the previous year's shoots resumes. It is not always possible to distinguish the suspension of growth according to the changing color of the perisarc in summer colonies.

Much less frequently examples of the growth arrest of an incompletely formed shoot module were observed. Then, upon the resumption of growth, various variants of anomalies are possible: expansion of the trunk {I-40(3)}; the formation of three hydranths or three branches sticking up; and the formation of the next internode not from the terminal surface of the growth apex but from its side.

The resumption of growth is possible not only at the top of the growth, but also at the hydranth. Hydranths in *D. pumila* usually exist for a limited time, after which they dissolve, and, in their place, under favorable conditions, new hydranths are formed. The process of the re-formation of the hydranth is well studied in hydroids of the genus Campanulariidae, since after the resorption of the hydranth, the hydrotheca falls off without closing the rudiment of the hydranth that appears in the previous location. This allows one to clearly see the end of the hydranth's life cycle and determine its duration, which is usually about one week [20–23].

In contrast to Campanulariidae, in all Sertulariidae, the hydrothecae do not fall off after the resorption of the hydranths. Therefore, it is not always easy to determine the formation of a new hydranth inside an old hydrotheca. Failures in the formation of secondary hydranths in old hydrotheca look like the following: an extension of the mouth of the hydrotheca {IV-3}, the formation of a new hydrotheca on the old hydrotheca {IV-1 (9)}, and the formation of a stolon from the mouth of the old hydrotheca {VII-20(10)}.

The second group of morphovariations includes all cases of an abnormal arrangement of normal structures, for example, lateral branches extending perpendicularly from the frontal plane, two lateral branches from one place under the hydrothecae, or a hydrotheca instead of a lateral branch.

The third group includes all the anomalies in the structure of the shoot, which are similar to shoots of other genera of the same family.

More often, there is a morphovariation of the asymmetric arrangement of the hydrothecae on the trunk or branches (i.e., hydrothecae in two rows with an offset relative to each other) {I-20(23)}. Additionally, the growth tip is, as it were, skewed, because hydrothecae are not formed simultaneously, but one before the other. This is similar to the typical arrangement of the hydrothecae in species from the genus *Abietinaria*.

A single-row arrangement of the hydrothecae occurs much less frequently, similar to the pattern found in species of the genus *Hydrallmania*. The similarity is enhanced by another feature of the structure of the hydrothecae, namely, an alternate orientation of the mouths of the hydrothecae in opposite directions. The distal parts of the hydrothecae, located in one row, are alternately slightly curved in a left–right–left pattern.

The second important feature of the third group of morphovariations is expressed in the repetition of the shape of the internode several times in a row, i.e., in the cycle of morphogenesis. This condition is optional. However, numerous cases of successive repetition of a form not characteristic of this species in shoots indicate the stability of such a repetitive morphogenesis. From such morphogenetic cycles, sections of the shoot are formed, consisting of completely perfect and slightly varying modules of a different form as compared with the characteristic shoot form of this species. Ordinary anomalies

are not capable of self-reproduction in a modular organism. They appear sporadically and end with growth arrest. However, morphovariations of the third group, during the process of completion of the shoot internode, reproduce the starting form to repeat the same alternative morphogenesis, i.e., they possess the necessary features of cyclic morphogenesis.

The fourth group includes all true deformities. As a rule, such deformities occur at the beginning of the formation of the shoot, after which it cannot grow further. Therefore, anomalies of the fourth group are difficult to detect; for this reason, it is necessary to carefully examine the shoots on the substrate without tearing them away from the stolons. Tiny abnormal shoots are hardly noticeable among large ones.

*3.2. Frequency of Occurrence of Morphovariations*

*Dynamena pumila* shoot anomalies differ in their frequency of occurrence (Figure 5).

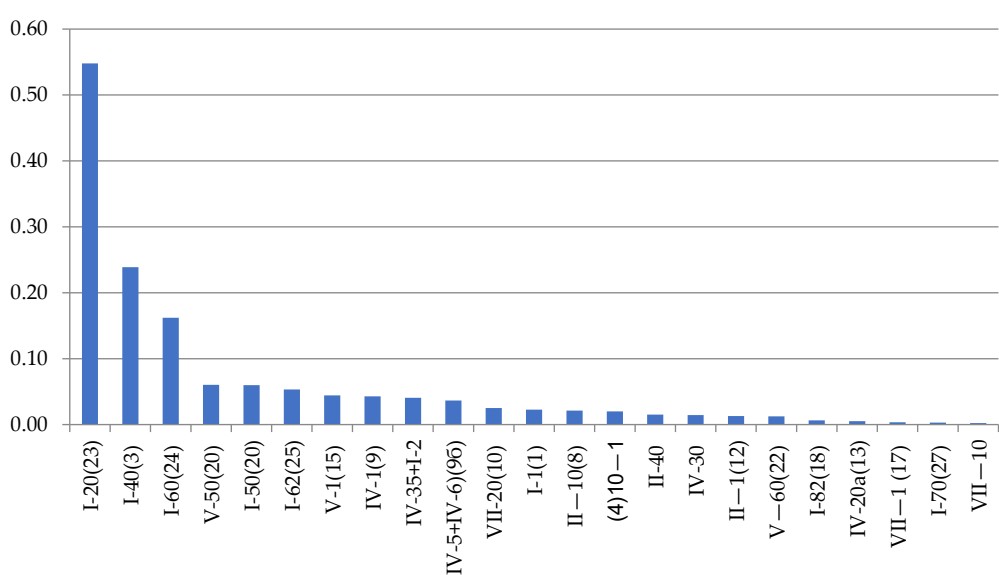

**Figure 5.** Frequency of morphovarieties in a sample of 230,700 shoot modules of *D. pumila* from the White Sea in descending order. Axes designations: "X"—anomaly indices (see Table 2; indices correspond to the full classification according to [9,16]) and "Y"—%.

The asymmetric position of the hydrothecae on the stem of the shoot absolutely prevails among morphovariations {I-20(23)}. Approximately two times less common is anomaly {I-40(3)}, i.e., the expansion of the shoot trunk. Even rarer is morphovariation {I-60(24)}, i.e., a single-row arrangement of the hydrothecae.

Table 2 shows not only the values of the number of each of the most common *D. pumila* anomalies, but also the frequency (%) of the anomaly in the total sample, i.e., the percentage of this morphovariation among all the modules in the sample. This gives a clearer idea of how rare deviations from the normal opposite-facing position of the hydrothecae in the shoot occur.

The cumulative frequency of the occurrence of all the *D. pumila* anomalies in a sample of 230,700 shoot modules was found to be 1.45%.

*3.3. Stems Anomalies in Colonial Hydroid Diphasia fallax (Johnston 1847)*

In total, 26 major morphovariations (Figure 6) were found among 21,537 *D. fallax* shoot modules, which occurred 99 times in total. Among them, 17 morphovariations were found to be more common than others (Table 3). In terms of morphology, they have a significant similarity with *D. pumila* morphovariations, but there are some differences.

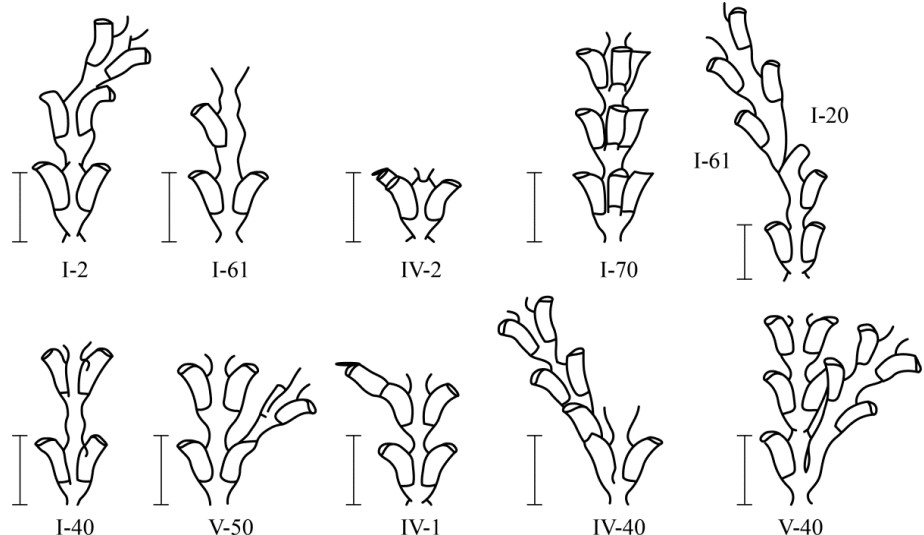

**Figure 6.** Anomalies in the structure of the shoots of *D. fallax*. See Table 3 for the anomaly numbers. The anomaly numbers correspond to the full classification (according to [9,16]). Scale bars = 1 mm.

**Table 3.** The 17 main anomalies of shoots of *D. fallax* in descending order of frequency of occurrence in a total sample of 21,737 shoot modules (White Sea).

| No. According to the Classification of Anomalies in *D. pumila* 1995 (1975) | Short Name for *D. fallax* Stem Anomalies | Sum of Anomalies in the Total Sample | Frequency of Occurrence of Morphovariations in the Total Sample (%) |
|---|---|---|---|
| I-1 | Stems bend | 38 | 0.175 |
| I-61 | Single row of hydrothecae due to the formation of only one hydranth in the internode | 29 | 0.133 |
| IV-2 | The mouth of the hydrotheca is built upon | 23 | 0.106 |
| I-70 | Three-row arrangement of hydrothecae | 21 | 0.097 |
| I-20 | Sequential arrangement of hydrothecae | 20 | 0.092 |
| I-40 | Thickening of the stem | 16 | 0.074 |
| IV-50 | Missing diaphragm in one or both hydrothecae | 17 | 0.078 |
| I-64 | The transition of a two-row arrangement of hydrothecae into a single-row, and then again into a two-row arrangement without intermediate stages with the formation of a giant hydrothecae | 16 | 0.074 |
| I-10 | Rotation of the plane of the colony around its own axis | 11 | 0.051 |
| V-30 | The lateral branch grows above the hydrotheca | 7 | 0.032 |
| I-50 | Long constriction between pairs of hydrothecae | 6 | 0.028 |
| V-1 | Lateral branch grows perpendicular to the frontal plane of the main shoot | 6 | 0.028 |
| IV-1 | Hydrotheca on hydrotheca | 3 | 0.014 |
| V-50 | Lateral branch grows from the mouth of the hydrotheca | 2 | 0.009 |
| IV-7 | Elongated hydrotheca with a displaced position of the diaphragm; the lower edge of the inner wall is separated from the hydrotheca | 1 | 0.005 |
| IV-40 | The development of the hydrotheca ends with the formation of an internode of a single-row stem | 1 | 0.005 |
| V-40 | Side branch instead of hydrotheca | 1 | 0.005 |
| | Total: | 218 | 1.00 |

Alternative morphotypes are highlighted in red.

The arrangement of the hydrothecae in an alternating pattern was found in *D. fallax* much less frequently than in *D. pumila* (Table 2), although almost every lateral branch extending from the shoot trunk had hydrothecae at different levels in the first module [16].

In *D. fallax*, more often than in *D. pumila*, elongated hydrothecae were formed due to the addition of an existing mouth {IV-2}, i.e., due to the re-formation of hydranths in place of the resolved ones (Figure 7). On the other hand, in *D. pumila*, elongated hydrothecae {IV-1(9)} were more often formed initially and not during repeated morphogenesis.

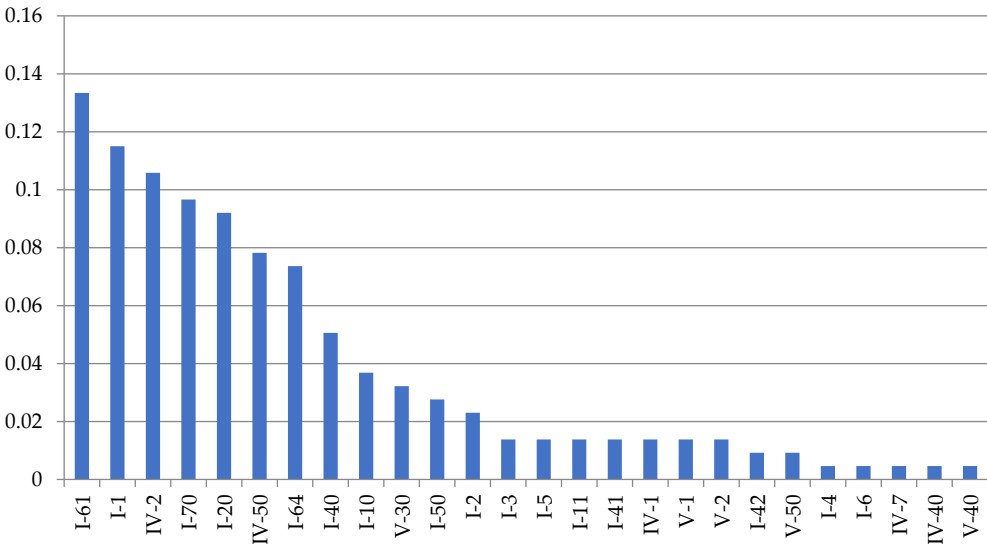

**Figure 7.** Frequency of anomalies (morphovariations) in a sample of 21,737 shoot modules of *D. fallax* from the White Sea in descending order. Axes designations: "X"—indices of morphovariations of shoots (see Table 2; indices correspond to the full classification according to [9,16]) and "Y"—%.

A single-row arrangement of the hydrothecae in *D. fallax* was more common than other morphovariations {I-61, I-64}, but differs somewhat from that of *D. pumila*. Usually, a single-row pattern occurs only in one, and less often two, escape modules, moving further into the usual double-row pattern.

A three-row arrangement {I-70(27a)} is much more common in *D. fallax* than in *D. pumila*.

It has been noted that the *D. fallax* hydrothecae sometimes lack a bottom, i.e., a diaphragm. This was not found in the other two species, although it cannot be ruled out that this anomaly was not noticed because an absence of a diaphragm is more difficult to observe than other anomalies.

The cumulative frequency of the occurrence of all the *D. fallax* anomalies in a sample of 21,737 shoot modules was found to be 0.84%.

*3.4. Return to Normal Morphogenesis*

If the appearance of aberrant modules does not lead to the cessation of shoot growth, then subsequent modules appearing after them on the same shoots are, as a rule, completely normal. This feature is important for the search for mechanisms of morphogenetic polyvariance.

Deviations from the norm in the development of the organism are associated, first of all, with mutations, genetic or somatic. The appearance of alternative phenotypes registered in a colony of hydroids is difficult to explain by mutations since, along with abnormal modules, there are always completely normal ones in the colony. Usually, in a colony, several tens, or even hundreds, of shoot tips grow at the same time. Whereas the morphogenesis may be different from the canonical one in one of them, in all the others, it may be no different from the norm. Since an entire colony is genetically identical, there is no reason to assume that the main cause of the appearance of abnormal modules is genetic mutation. Moreover, somatic mutations cannot easily be described as the main mechanism of morphogenetic polyvariance. In the apical part of the growth apex, precisely in the zone of morphogenetic transformations, mitoses are absent. The zone of mitotic

activity, i.e., cell proliferation, is located at the base of the growth apex. Even if we assume that a somatic mutation has occurred in any cell of the growth apex and, as a result of its multiplication, affected the subsequent process of morphogenesis, the rudiment of the next module, isolated on the apical surface of the growth apex, should thus consist of the same mutant cells, which means that the next module must be aberrant. However, this is very rare.

Thus, the return to normal morphogenesis after the realization of a different morphogenetic program should testify in favor of non mutational causes of morphogenetic polyvariance, thereby supporting the hypothesis of an epigenetic origin for deviations from the norm in the structure of modules.

### 3.5. Anomalies in the Colonial Hydroid Abietinaria abietina (Linnaeus 1758)

In total, 39 main morphovariations were found among 20,395 *A. abietina* shoot modules (Figures 8 and 9). Among them, four morphovariations are basic. The frequency of occurrence in each of them was 0.20–0.21% (of the sum of the internodes of the entire sample) (Table 4). All of them are represented by deviations from the normal form of the hydrothecae. In the next group of anomalies, with a frequency of occurrence from 0.08 to 0.13%, deviations from the norm in the orientation of the trunk or branch of the shoot predominate. Rarer anomalies with a frequency of occurrence of <0.04% also affect the morphogenesis of hydrants. Among the anomalies, an elongated mouth of the hydrotheca was found, that was curved down {IV-31} or not turned away from the stem of the shoot but upwards and parallel to the shoot {IV-31a}. Hydrothecae can be tubular or barrel shaped. Hydrothecae have two, or even three, diaphragms (probably a consequence of regeneration after resorption of the hydranth); it is not uncommon for the mouth of the hydrotheca to be thickened (like a rim) {I-47}. The hydrothecae were underdeveloped {IV-37}, which is very rare in *D. pumila*.

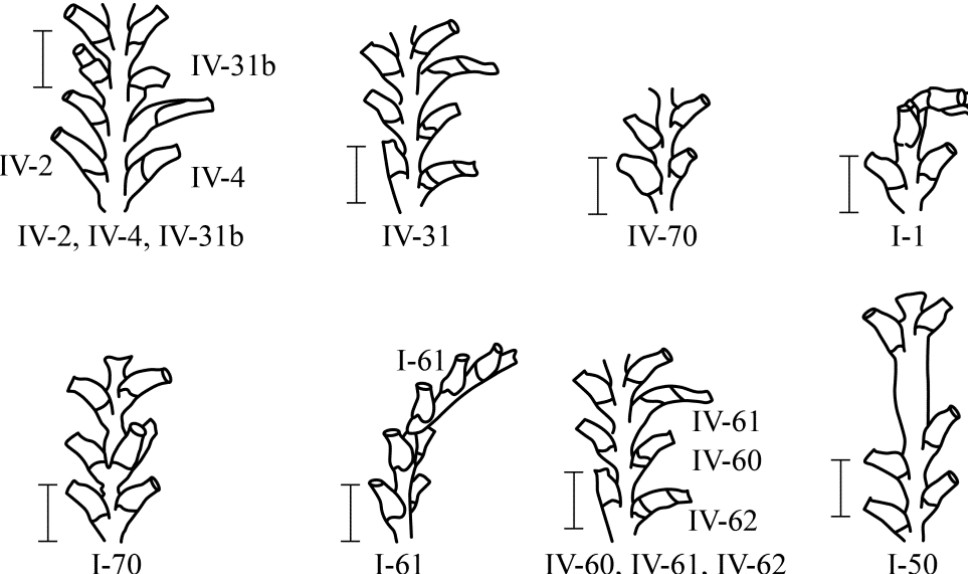

**Figure 8.** The main anomalies of *A. abietina* shoots, the occurrence of which is presumably due to errors in the differentiation of the hydranths and during their re-formation. The anomaly numbers correspond to the complete classification (according to [9,16]). Scale bars = 1 mm.

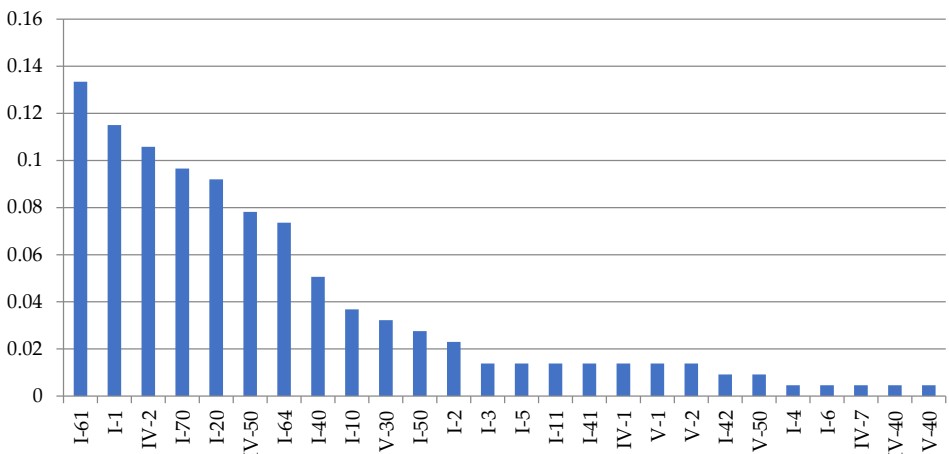

**Figure 9.** The frequency of occurrence of the main morphovariations of *A. abietina* shoots in descending order from a sample of 20,457 shoot internodes. The names of anomalies can be found in Table 4. The designations and names of the anomalies correspond to the full classification (according to [9,16]).

**Table 4.** The 27 main shootanomalies of *A. abietina* in descending order of frequency of occurrence in the total sample of 20,395 shoot modules (White Sea).

| Anomaly Classification No. for *D. pumila* (1995) | Anomalies of Shoot Internodes of *A. abietina* | The Sum of Anomalies in the Total Sample | Frequency of Occurrence of Anomalies in the Total Sample (%) |
|---|---|---|---|
| IV-2 | Hydrothecae with built-in mouth | 43 | 0.21 |
| IV-4 | Extended high-diaphragm hydrothecae | 43 | 0.21 |
| IV-31a | The mouth of the hydrotheca faces the axis of the shoot | 42 | 0.21 |
| I-47 | Thickening in the form of a collar of theca (cup-shaped rim) | 40 | 0.20 |
| I-11 | Rotation of the axis by 90 $^\circ$ | 26 | 0.13 |
| I-2 | Trunk curvature in the frontal plane with internode deformation | 23 | 0.11 |
| IV-37 | Underdeveloped | 18 | 0.09 |
| I-10 | Rotation of the axis by less than 90 $^\circ$ | 16 | 0.08 |
| V-1 | Lateral branch perpendicular to the frontal plane of the shoot | 16 | 0.08 |
| IV-31 | The mouth of the hydrothecae is bent outward and down | 9 | 0.04 |
| I-61a | Hydrotheca located outside the frontal plane | 8 | 0.04 |
| IV-70 | Barrel-shaped hydrotheca | 7 | 0.03 |
| I-61 | <span style="color:red">Single-row arrangement of hydrothecae (only in a single internode of one hydrotheca)</span> | 7 | 0.03 |
| IV-60 | Two diaphragms in the hydrotheca | 6 | 0.03 |
| I-40 (3) | Thickened section of the trunk | 4 | 0.02 |
| I-70 | <span style="color:red">Triad hydrothecae (three-row arrangement)</span> | 4 | 0.02 |
| I-50 | Extended section of the stem without hydrothecae | 2 | 0.01 |
| I-30 | Closed pair of hydrothecae (stem behind) | 2 | 0.01 |
| IV-39 | Hydrotheca was formed in the sinus of the previous hydrotheca | 2 | 0.01 |
| IV-1(9) | Hydrotheca on hydrotheca | 1 | 0.00 |
| II-11 | The upper pair of hydrothecae is elongated and tubular without a growth tip between them with high diaphragm | 1 | 0.00 |
| IV-2a | Hydrotheca with a narrower built-on part | 1 | 0.00 |
| IV-61 | Tubular hydrotheca with a high-positioned double diaphragm | 1 | 0.00 |
| IV-62 | Tubular hydrotheca with a high-positioned triple diaphragm | 1 | 0.00 |
| V-30 | Lateral branch from the sinus of the hydrotheca | 1 | 0.00 |
| V-31 | Lateral branch from the sinus of another lateral branch | 1 | 0.00 |

Alternative morphotypes are highlighted in red.

*A. abietina* is characterized by an alternate arrangement of the hydrothecae. This species-specific feature is due to the asymmetry of the growth apex (Figure 10), in which the bias in one direction changes to the opposite direction whenever the next shoot internode is formed. Therefore, asymmetry is deeply rooted in the shoot morphogenesis. Moreover, lateral shoot initiation in *A. abietina* occurs during the formation of the apical module, i.e., lateral shoots are laid at the top of the growth in the form of a lateral bud on a still-unformed hydranth. In *D. pumila* and *D. fallax*, the lateral branches are laid much later than the formation of the upper internode. They are formed in the form of a lateral kidney below the hydrotheca diaphragm, i.e., on the leg of the formed hydranth (Figure 1). Strictly opposite, single-row and three-row arrangements of hydranths are weakly expressed, rare, and do not repeat cycle after cycle during morphogenesis. However, *A. abietina* may have these morphotypes.

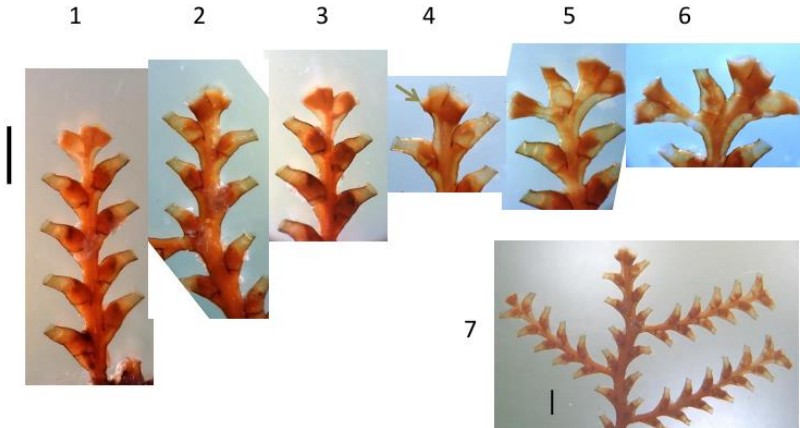

**Figure 10.** Photographs of *A. abietina* shoots at successive stages of the morphogenetic cycle. 1—The rudiment of the hydranth on the left and the rudiment of the trunk and the hydranth, still undivided, on the right; 3—the rudiment of the hydranth on the right and the rudiment of the trunk and the hydranth at the beginning of dismemberment on the left; 4—the rudiment of the lateral branch is indicated by an arrow; 5—a side branch, a hydranth, and the beginnings of two hydranths at different levels and a trunk between them; 7—general view of the distal part of the *A. abietina* shoot and regular arrangement of lateral branches. Scale bars = 1 mm.

## 4. Discussion

Intraspecific variability, as a rule, is studied in populations, i.e., in genetically non-identical objects. Therefore, researchers explain any differences between representatives of the same species according to genetic differences, sexual dimorphism, age-related changes, or the body's response to external influences. Thus far, intraspecific variability has been studied mainly on unitary organisms. The variability of colonial organisms has always been studied in the same way as that of solitary organisms, i.e., by using population samples [24–27].

In contrast to unitary organisms, in modular organisms, which includes hydroids, corals, bryozoans, and other colonial invertebrates, as well as plants and fungi, variability can manifest itself within one organism in modules of the same type. There are also separate studies of intraorganismal variability in plants [28–32]. Such variability cannot be explained either by sexual dimorphism or by differences in life cycle stages. It is assumed that somatic plasticity of genetic regulation is manifested in module variations [33], i.e., this is a morphogenetic polyvariance described in more detail in hydroids [9]. Usually, morphogenetic polyvariance in hydroids, using the example of two phenotypes, polypoid and medusa, is explained not only by a regular change in the course of ontogenesis, but also by the influence of environmental factors [34].

In colonial organisms, it is easy to test assumptions regarding the influence of environmental factors on morphology. For this purpose, it suffices to compare the observed

deviation from the norm in the structure of the zooids or shoots against other zooids or shoots that formed simultaneously. In cases where morphological changes arise under the influence of external influences on the entire colonial organism (for example, temperature, salinity, or the composition of the aquatic environment), one should expect a simultaneous reaction of morphogenesis in all parts of the colonial whole that are currently being formed. If morphological changes are single and are not repeated in other similar parts, e.g., shoots or zooids, then another reason for the appearance of anomalies should be sought. The synchronism of the occurrence of abnormal morphotypes can be easily established even in fixed samples by determining the distance between the anomaly and the growth tip. The growth of branches is localized at their tips and the modules are formed in a certain time, for example, in *D. pumila* hydroids, this occurs in one day (at 14–16 °C); therefore, the detection of anomalous modules at the same distance from the growth tips of the shoots would give reason to believe that deviations from the norm in morphogenesis occurred under the influence of some external factor. However, as a rule, anomalies are located at different distances from the growth tips of the shoots.

Large shoots of the three studied hydroid species were always branched and included dozens of internodes. This made it possible to judge the degree of synchronism in the formation of anomalies in the branches of the shoot. Nevertheless, not a single example of the simultaneous formation of abnormal internodes in the branches of large shoots of *D. pumila*, *D. fallax,* and *A. abietina* was found. By simultaneous is meant an anomalous internode that is in an equidistant position from the top of the branch. Moreover, the frequency of anomalies is usually so low that it almost impossible to detect single anomalies in every branched shoot. Less common are shoots with several different morphovariations of their internodes. Even less common are shoots with the same morphovariation but, as a rule, at different levels of the trunk and branches. There are no examples of repeated anomalous internodes located equidistant from the tops of the trunk and shoot branches. This fact makes it possible to doubt the assumption regarding the role of ordinary external environmental factors in the formation of anomalous shoot internodes. More likely is the hypothesis of the independent (endogenous) appearance of anomalies, or the assumption of the presence of some thus far unknown local environmental factors that affect only some branches of the shoots, but not all at once.

When using the term "anomalous internode (or module)", it should be noted that anomalies in the structure of internodes can only be qualified as an abnormality or as a deformity with some degree of conventionality. Most anomalies are morphologically stable. Their diversity is small, and most are represented by variations of the forms that are usual for this species, for example, extensions of the trunk similar to the top of the growth, one hydranth at the top instead of the next internode, a lateral branch from a hydrotheca, an alternate arrangement of the hydrothecae, or a one-row and three-row arrangement of the hydrothecae.

When comparing morphovariations in different types of hydroids, one would expect the manifestation of species specificity not only in the normal structure of the shoot, but also in deviations from the norm. However, the results of this study indicate the opposite—most morphovariations are similar in *D. pumila, D. fallax*, and *A. abietina*. Differences between species are mainly manifested not in the appearance of some unique morphovariations, but in the frequency of occurrence of the same deviations from the norm. For example, curvature of the hydrothecae with the mouth pointing upwards is among the ten predominant morphovariations in *A. abietina*, whereas in *D. pumila* and *D. fallax,* this is extremely rare. On the other hand, in *D. fallax,* it is not uncommon for a three-row arrangement of the hydrothecae; in *A. abietina,* it is several times less common, and in *D. pumila,* this is very rare (Table 5). The similarity of the set of morphovariations in the three species indicates a common tendency for several genera of the same family to form certain phenotypes that differ from the main species-specific phenotype.

**Table 5.** Relative frequencies of occurrence of the dominant morphovariations in three species of colonial hydroids: *Dynamena pumila*, *Abietinaria abietina*, and *Diphasia fallax* (percentage of occurrence of each morphovariation among the total number of all morphovariations in the sample).

| Line Name | No. of Morphovariations | % of the Indicated Morphovariation Occurrence among the Total Number of All Anomalous Shoot Internodes in the Sample | | |
|---|---|---|---|---|
| | | *Dynamena pumila* | *Abietinaria abietina* | *Diphasia fallax* |
| *Samples(number of internodes)* | | 230,700 | 20,395 | 21,537 |
| Number of all anomalies of the shoot internodes in the sample | Together | 3347 | 325 | 218 |
| Cumulative frequency of occurrence of anomalies | Together | 1.45 | 1.59 | 1 |
| Hydrothecae in two rows with an offset relative to each other | I-20(23) | 37.7 | normal | 9.18 |
| Trunk thickening | I-40(3) | 16.46 | 1.23 | 7.34 |
| Single-row arrangement of hydrothecae | I-60(24) | 11.17 | — | 0 |
| One of two hydranths in an internode | I-61 | rare | 2.15 | 13.3 |
| Double-row→single-row→double-row hydrothecae arrangement | I-64 | rare | — | 7.34 |
| Long constriction between internodes | I-50(20) | 4.12 | 0.62 | 2.75 |
| The lateral branch of the shoot grows from the mouth of the hydrotheca | V-50(11) | 4.15 | — | 0.92 |
| The lateral branch of the shoot grows perpendicular to the frontal plane | V-1(15) | 3.05 | 4.92 | 2.75 |
| Hydrotheca from the mouth of the hydrotheca | IV-1(9) | 2.96 | 0.31 | 1.34 |
| Elongated hydrotheca | IV-2 | rare | 23.46 | 10.55 |
| Shoot trunk bend | I-1(1) | 1.55 | 7.08 | 17.43 |
| Absence of hydrotheca diaphragm | IV-50 | — | — | 7.8 |
| Three-row arrangement of hydrothecae | I-70(27a) | 0.21 | 1.23 | 9.63 |
| The mouth of the hydrotheca faces the trunk | IV-31 | — | 12.92 | — |

Among all the alternative phenotypes, four are distinguished, the morphogenesis of which is different, namely:

- Opposite symmetrical arrangement of the hydrothecae in two rows;
- Asymmetric (alternate) arrangement of the hydrothecae in two rows;
- Single-row arrangement of the hydrothecae;
- Three-row arrangement of hydrothecae.

An opposite symmetrical arrangement of the hydrothecae in two rows is formed with a gradual flattened expansion of the growth apex that initially has a hemispherical shape. This variant of morphogenesis was studied using the example of *D. pumila*. It was first described in detail by L.V. Belousov [35,36]. In the process of a flattened expansion, the shoot tip is initially monolithic, and, after reaching a certain proportion of width to its height, one rudiment is divided into three, which is achieved due to the protrusion of the coenosarc wall and the formation of two vertical semi-septa of the perisarc (Figure 2).

Semi-partitions dissect only the upper part of the growth apex. It was found that the invagination of the walls of the growth apex does not occur from top to bottom as previously assumed, but from its sides [37]. Further growth of the shoot tip is accompanied by the expansion and bending of the lateral lobes, which become two hydranths in the hydrothecae. If the entire apex was initially a single whole, which can be judged by its pulsations, after dividing itself into three rudiments, the two lateral apexes gradually become less dependent on the central one, which can also be seen from the pulsations. This is especially evident after the completion of the formation of hydranths (according to our own observations).

Lateral branches with an opposite symmetrical arrangement of the hydrothecae in *D. pumila* and *D. fallax* appear later than the formation of internodes. They are laid as independent buds below the hydrothecae in the frontal plane of the shoot and grow at a certain angle to the axis of the shoot. The planes of the lateral branches coincide with the plane of the shoot trunk. Since the axis of the lateral branch forms an acute angle with respect to the distal part of the shoot trunk, the tip of the branch may be slightly skewed so that the first pair of hydrothecae does not look quite symmetrical. This is reminiscent of the asymmetry of the alternating arrangement of the hydrothecae. However, such an asymmetry normally does not affect the shape of the subsequent internodes of the lateral branches since constrictions are always formed, or formed at least once, between successive interstices. In a cross section, the constriction is round, i.e., the asymmetry of the previous internode disappears. A new internode always starts with a symmetrical hemisphere.

The shoot morphogenesis of *D. fallax* has never been studied in any detail [38]. Thus, this can be judged only indirectly according to the shape of the growth tips that occur at different stages of development. From this point of view, the shoot morphogenesis of *D. fallax* in all the main locations coincides with the morphogenesis of *D. pumila* described above [39], except for its tendency to form stolon-like processes on the tops of large shoots.

The morphogenesis of *A. abietina* seems to be similar to that of the two species described above at first glance. However, the difference between them is important. In *A. abietina*, the lateral branches are not formed independently on the stem of the shoot but are programmed during the formation of the next internode of the shoot (Figure 10). The shoot apex is initially asymmetric (skewed). Like *D. pumila* and *D. fallax*, the tip of *A. abietina* is flattened, but its two "shoulders" are at different levels. The laying of transverse furrows does not occur simultaneously, but alternately. First, two different-sized transverse furrows are laid from the side of the towering "shoulder" of the top of the shoot. This results in the formation of a hydranth and a side branch below it. The third rudiment from the side of the opposite "shoulder" is larger in size than each of the two into which the towering "shoulder" is divided. Having increased in size as it grows, the third rudiment is subdivided, in turn, by two different-sized furrows into a hydranth and a side branch. This asymmetric laying of two furrows determines the non-opposite position of the hydranths and the regular laying of lateral branches.

Another variant of the single-row arrangement of hydranths on a shoot is well known in the example of the genus *Hydrallmania*. In *H. falcata*, in the lateral branches of the first and second orders, all the hydrothecae are located in one row. At the same time, their mouths are alternately turned in opposite directions (Figure 11). The morphogenesis of this pattern is reduced to an asymmetric division of the growth apex by oblique furrows (semi-septae) and is spatially more complex than that in *A. abietina*. The furrows do not run across or along the top, but run obliquely. With each division into two parts, the larger part is divided into two smaller rudiments; the smaller part of the apex initially becomes the main one, grows, and soon also divides into two parts. Obliquely positioned dividing furrows predetermine the alternating orientation of the hydrotheca axes in two directions. This scheme of differentiation of the shoot growth apex is complicated by the regular laying of lateral branches on it, which occurs in an alternating arrangement on its opposite lateral sides. Essentially, in this variant of morphogenesis, the principle of repeated subdivision of the growth apex into unequal primordia and lateral branches remains the same as in *A. abietina*.

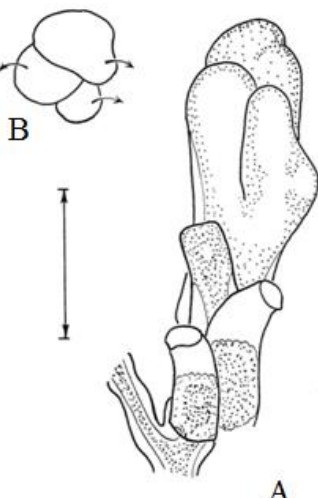

**Figure 11.** The distal part and growth tip of the lateral branch of the shoot of *Hydrallmania falcata* in the frontal plane (**A**) and above (**B**). Arrows indicate subsequent turns of the primordia during their growth. Scale bar = 0.5 mm. (Adapted from [40]).

In young shoots of *H. falcata*, the hydranths are located not in one row, but in two opposite-facing alternating ones, and they occur strictly according to the principle of sympodial branching (Figure 12) [1,39]. Consequently, two patterns of morphogenesis can be realized within one colonial organism. The combination of a single-row pattern with a sympodial one confirms the hypothesis that the multi-row pattern in the order Leptothecata originates from the sympodial type of shoot structure [41]. The multi-row arrangement of the hydranths could arise with an ever-earlier laying of the next internode. Usually, the next internode is laid on the previous one after the differentiation of the hydranth is completed on it. This normally occurs in many members of the Campanulariidae family, for example, *Gonothyraea loveni* (Allman, 1859) and *Laomedea flexuosa* Alder, 1857 [42]. It has long been noticed that this rule is violated in some species from the same family. The stage of laying the next internode in *Obelia geniculata* (Linnaeus, 1758) may come before the completion of the development of the previous hydranth [43]. If this advance in the laying of the next internode progresses evolutionarily, then the rudiments of the hydranth and the next internode should merge. As soon as the hydranth begins to be determined, the bud of the next hydranth is immediately formed on it, and so on. This reduces the length of the internodes, and the hydranths are formed continuously. The coenosarc tube and the hydranth stem gradually shrink during this evolution, turning into a fragment of the coenosarc of a more complex shape, and the hydrotheca ceases to be goblet shaped, with one of its sides remaining inseparable from the internode [41]. All of these are signs of heterochrony—one of the main "engines" of the evolutionary process and the biological mechanism of species variability—the generation of diversity based on the same genotype [44].

The evolutionary transformation of the sympodial structure into a multi-row structure (monopodial with an apical growth zone) highlights a significant complication of shoot morphogenesis. A simple version of morphogenesis consists of a unitary undivided top of a hemispherical shape, which remains so until the last stage of hydranth differentiation, as occurs in many hydroids with simply arranged colonies. A more complicated version consists of an asymmetric top, subdivided into parts before the completion of the formation of the hydranth. This evolutionary transition from a simple and radially symmetrical pattern to a more complex one could well be accompanied by failures in morphogenesis. The instability of morphogenesis is manifested in the polyvariance of the morphovariations of the shoot internode. The diversity of morphovariants is realized in the diversity of multi-row hydroids that was established long before the morphological stabilization of the characteristics of species, genera, and even families.

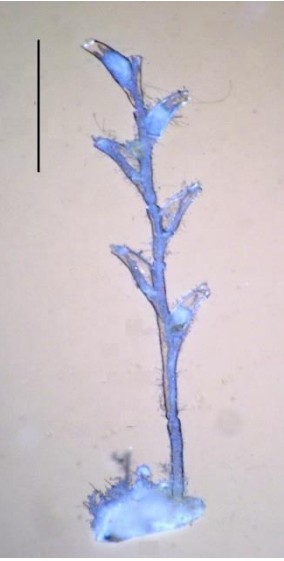

**Figure 12.** Symbodial pattern of a small primary shoot of *Hydrallmania falcata*. Scale bar = 1mm. (Photo courtesy of I.A. Kosevich).

Unlike the multi-row hydroids, the sympodial ones did not show any significant anomalies in the structure of the shoots. There are still no dedicated studies on this topic, but according to our observations the only anomaly in the structure of sympodial shoots is the dichotomous branching of the internode in *G. loveni* and *L. flexuosa*.

Thus, the single-row and double-row (both symmetrical and asymmetric) arrangement of the hydranths are the products of three different morphogenetic patterns that are evolutionarily related to each other.

The three-row arrangement of the hydranths was found in all three species, but with different frequencies of occurrence. In *D. pumila*, the three-row arrangement is very rare, but in *D. fallax*, it is not as rare, and can even be considered relatively common (Table 5). The morphogenesis of the three-row formation of the hydranths has not been studied at all [45]. According to the shape of the growth tips during the process of growth, it can be argued that, firstly, they do not become flattened, and, secondly, they are divided at a certain stage of morphogenesis into four primordia: one in the center and three around it. The angle between the outer three primordia is normally 120°. In the rare cases when this anomaly does occur, the three-row form of the internode is repeated in the shoot several times in a row, and, in subsequent modules of the same shoot, the usual two-row structure of the shoot may resume.

Therefore, the three-row arrangement of the hydranths is the result of a special morphogenesis, like three other morphovariations: one single-row arrangement and two double-row arrangements. Therefore, it is convenient to classify all of these as morphotypes, in contrast to other deviations from normal morphogenesis.

The remaining morphovariations, judging by their form and place of appearance, are most likely variants of normal morphogenesis. Some of them fit the notion of "mistakes of place" in the implementation of the genetic program, for example, a stolon from a hydrotheca or at the top of a shoot, a gonotheca from a hydrotheca, or a side branch from a gonotheca.

Other morphovariations are, apparently, a consequence of a temporary stoppage of shoot growth when the tip freezes without completing its full development with the formation of an internode. The thinnest perisarc on the terminal (upper) part of the apex remains extensible with constant growth, and, when growth stops, it polymerizes, becomes thicker, and is no longer elastic. Under favorable conditions, growth may continue, but only from the tip of the shoot. Sometimes growth resumes from above the top, and sometimes from the side. In any case, the germ of the next internode breaks through where the perisarc

is dissolved by a new growth bud of the coenosarc. Thus, there are various curvatures of the escape. A special variant of the curvature of the shoot is associated with the asymmetric formation of the hydranths when one of them is skipped, i.e., when two hydrates are formed in a row on one side and none on the opposite side.

If limited deviations from the normal phenotype can be interpreted as ordinary intraspecific variability, then the four morphotypes described above do not look like deviations from the norm, but rather are alternative phenotypes: harmonious, stable, and, most importantly, arising according to a completely different morphogenetic program. These are, rather, "reserve" morphogenesis of the family—programs of individual development, completely ready to become the main ones in the ontogeny of the species.

In hydroids of the genus *Sertularia*, morphogenetic polyvariability manifests itself regularly, not randomly. On the shoot trunk of *S. mirabilis* (Verrill, 1873), the hydrothecae are located in two opposite rows, and in the branches of the first order, the hydrothecae are arranged in six rows, and occasionally in four rows [11,38]. The number of rows of the hydrothecae, as it was found, depends on the diameter of the branch, and as the diameter increases, the rows of the hydrothecae also become larger [46]. In young primary shoots that have grown from settled planula, a pair of hydranths is usually immediately formed; however, they are not strictly opposite, but have some shifts, i.e., they look like an alternate bookmark (judging by the position of the hydrotheca diaphragms). Sometimes, only one hydrotheca is initially laid instead of a pair [46]. This is an example of genetically fixed options for the location of hydrothecae on the trunk and branches.

In the three species studied here that have a two-sided arrangement of the hydrothecae, the options for the number of rows of the hydrothecae other than the two-row arrangement (i.e., a single row or three rows) are "illegal", i.e., they should normally not be formed. The deviations from the norm described here are not regular and are most likely random. However, it is precisely these morphotypes that are not accounted for by normal morphogenesis, suggesting that, in addition to the regular versions of morphogenesis, there may be hidden ones. These hidden or "dormant" morphotypes are provided, as can be observed, by the genetic program. They are perfect in form, and the hydranths are able to catch prey and digest food. Although such morphotypes are not in demand, they remain "in stock".

The generally accepted mechanisms for the generation of variability in populations, such as mutations, horizontal gene transfer, meiosis, and the adaptive immune system [47], primarily cause changes in the genotype. However, modular organisms, such as hydroids, exhibit phenotypic variability within one organism based on heterochrony occurring not at the population level but within the same organism. According to the proposed division of developmental plasticity into two forms: developmental conversion and phenotypic modulation [48], morphogenetic polyvariance should rather be considered to be developmental conversion. This applies to morphotypes at least.

In colonial organisms, alternative morphogenesis or alternative morphotype modules are protected from natural selection by the prevailing conventional modules, the phenotype of which is strictly species specific. If the prevailing variant of the structure of the modules is ecologically perfect, then a small number of modules of a different structure will not affect the final result of natural selection in any way. This phenomenon can also be considered from the standpoint of the possibility of switching from one phenotype to another, including the possibility of returning to the previous phenotype, which has been discussed in the scientific literature. However, it is not necessary to link this phenomenon with the influence of environmental factors.

The rate of evolution in this case should be determined by the rate of change in the frequency of occurrence of minor morphotypes. It is not yet understood how the frequency of the occurrence of anomalies is regulated. Thus far, there are no studies on this topic. However, we know, owing to the study of dozens of samples of *D. pumila* shoots, that the frequency of the occurrence of morphovariations does not vary within a wide range, but within rather narrow ranges. This comparison of the degree of morphogenetic polyvariance in three species of hydroids made it possible to reveal significant differences

in the frequency of the occurrence of some morphovariations in the compared species. Moreover, it is clear that, in the three species, the morphological deviations from the norm are the same, i.e., they must be part of the characteristics of the family or subfamily.

## 5. Conclusions

In modular organisms, which includes colonial hydroids, morphogenetic polyvariance was found, which manifests itself in differences in the structure of stereotypical shoot internodes.

Most morphovariations of the shoot internodes are common in the three compared species of hydroids from different genera of the family Sertulariidae. However, their predominant morphovariations were found to be different.

The frequency of the occurrence of the dominant morphovariations was low. It was found to be in the range of 0.01–0.5% of the number of shoot internodes in the sample.

We found several unique morphovariations specific to only one of the three species. However, there were few of them; most of the morphovariations in the three species were homologous.

The data presented confirm the hypothesis that morphovariations are not the product of mutations, but are due to normal variability that is genetically preserved in each organism, which is expressed in the possibility of the implementation of several, rather than one, phenotypes in each organism.

**Funding:** This study was performed within the framework of the scientific project of the state task of Lomonosov Moscow State University no. 121032300118-0. Supported by RFBR grant # 122012700112-5 2022-2023.

**Institutional Review Board Statement:** Not applicable.

**Informed Consent Statement:** Not applicable.

**Data Availability Statement:** Not applicable.

**Acknowledgments:** The author thanks Elena M. Mayer, Rivva Y. Margulis, Boris V. Yuzenkov, Yuri B. Burykin, Anna B. Epelbaum, Tanya Orlova (Mayorova), and Anna Sergeeva for their help in processing the material and Igor A. Kosevich for providing the photo and discussion of the obtained results, as well as Thomas C.G. Bosch (Zoological Institute Christian-Albrechts-University Kiel) for valuable advice during the discussion of the study results.

**Conflicts of Interest:** The author declares no conflict of interest.

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
