# Peer review of "Hidden Morphotypes and Homologous Series in Phenotype Variations in the Colonial Hydroids Dynamena pumila, Diphasia fallax, and Abietinaria abietina (Hydrozoa, Leptothecata)"

_2673-6500, doi:10.3390/taxonomy2030027_

Round 1

Reviewer 1 Report

This work provides valuable data and knowledge about the phenotypic variations present in modular organisms such as colonial hydroids, proposing some hypotheses that are resolved almost satisfactorily.

The applied methodology seems correct and answers the three questions raised in the work. In addition, the specimens sampled for this study exceed the number recommended by the specific bibliography. Adding two new species to the study enriches the knowledge on how the different phenotypes in different genera of the same family vary, in frequency of appearance, expanding the knowledge cited in the reference literature. The schemes are appropriate and help in the understanding of the different phenotypes, as well as the corresponding explanation of how each other develops.

Although not clearly stated by the author, I think it would be interesting to hypothesize whether these specific abnormal growths might show "scarring" as a result of injury caused by external factors (e.g. predation, strong wave energy causing rupture of colonies, etc.), and/or even speculate if this could be , verified experimentally.

Among minor things, throughout the manuscript the author refers to "we, ours, or us", even though he is the sole author of the work (Lines: 107, 141, 186, 198, 261, 406, 462, 471, 473, 475, 637, and perhaps another). I understand that the author had the support of several assistants, as mentioned in the acknowledgments, but I think it would be more appropriate to write in the third person throughout the entire manuscript. 

Also, some scientific names are not italicized and need to be corrected (Lines: 333, 336, 339, 342, 343, 344, 347, 349, 351, 352, 355, 356, 357, 361, 588, and perhaps another). 

The word "in" is duplicated (Line: 351).

For the above, I recommend that the article be accepted for publication after minor corrections.

Reviewer 2 Report

The manuscript described hidden morphotypes and homologous series in the hydroids Dynamena pumila, Diphasia fallax and Abietinaria abietina. I think that the manuscript suitable after some minor corrections. Please see attached my checked manuscript.
